# Practical Management of Biosimilar Use in Inflammatory Bowel Disease (IBD): A Global Survey and an International Delphi Consensus

**DOI:** 10.3390/jcm12196350

**Published:** 2023-10-03

**Authors:** Ferdinando D’Amico, Virginia Solitano, Fernando Magro, Pablo A. Olivera, Jonas Halfvarson, David Rubin, Axel Dignass, Sameer Al Awadhi, Taku Kobayashi, Natália Sousa Freitas Queiroz, Marta Calvo, Paulo Gustavo Kotze, Subrata Ghosh, Laurent Peyrin-Biroulet, Silvio Danese

**Affiliations:** 1Gastroenterology and Endoscopy, IRCCS Ospedale San Raffaele and Vita-Salute San Raffaele University, 20132 Milan, Italy; damico.ferdinando@hsr.it; 2Department of Biomedical Sciences, Humanitas University, Pieve Emanuele, 20072 Milan, Italy; virginia.solitano@humanitas.it; 3Department of Medicine, Division of Gastroenterology, Schulich School of Medicine, Western University, London, ON N6A 3K7, Canada; 4CINTESIS@RISE, Faculty of Medicine of the University of Porto, 4200-319 Porto, Portugal; fernandomagro65@gmail.com; 5Gastroenterology Department, CEMIC, Buenos Aires C1425ASS, Argentina; pablo.olivera@mail.utoronto.ca; 6Zane Cohen Centre for Digestive Diseases-Lunenfeld-Tanenbaum Research Institute-Sinai Health System-Gastroenterology, Toronto, ON M5T 3L9, Canada; 7Department of Gastroenterology, Faculty of Medicine and Health, Örebro University, SE-701 82 Örebro, Sweden; jonas.halfvarsson@regionorebrolan.se; 8The University of Chicago Medicine Inflammatory Bowel Disease Center, University of Chicago Medicine, Chicago, IL 60637, USA; drubin@bsd.uchicago.edu; 9Department of Medicine I, Agaplesion Markus Hospital, Goethe University, 60323 Frankfurt am Main, Germany; axel.dignass@agaplesion.de; 10Digestive Diseases Unit, Rashid Hospital, Dubai P.O. Box 4545, United Arab Emirates; saalawadhi@dha.gov.ae; 11Center for Advanced IBD Research and Treatment, Kitasato University Kitasato Institute Hospital, Tokyo 108-8642, Japan; kobataku@insti.kitasato-u.ac.jp; 12Health Sciences Graduate Program, School of Medicine, Pontifícia Universidade Católica do Paraná (PUCPR), Curitiba 80215-901, Brazil; nataliasfqueiroz@gmail.com; 13IBD Unit, Gastroenterology, Hospital Universitario Puerta de Hierro Majalahonda, 28222 Madrid, Spain; marta.calvo@salud.madrid.org; 14Colorectal Surgery Unit, Hospital Universitário Cajuru, Pontifícia Universidade Católica do Paraná (PUCPR), Curitiba 80215-901, Brazil; pgkotze@hotmail.com; 15APC Microbiome Ireland, College of Medicine and Health, University College Cork, T12 E138 Cork, Ireland; sughosh@ymail.com; 16Department of Gastroenterology, Nancy University Hospital, F-54500 Vandœuvre-lès-Nancy, France; peyrinbiroulet@gmail.com; 17Inserm, NGERE, University of Lorraine, F-54000 Vandœuvre-lès-Nancy, France; 18INFINY Institute, Nancy University Hospital, F-54500 Vandœuvre-lès-Nancy, France; 19FHU-CURE, Nancy University Hospital, F-54500 Vandœuvre-lès-Nancy, France; 20Groupe Hospitalier Privé Ambroise Paré-Hartmann, Paris IBD Center, F-92200 Neuilly sur Seine, France; 21Division of Gastroenterology and Hepatology, McGill University Health Centre, Montreal, QC H4A 3J1, Canada

**Keywords:** biosimilars, bio-originators, biologic drug, IBD, ulcerative colitis, Crohn’s disease, interchangeability, economics

## Abstract

As the patents for biologic originator drugs expire, biosimilars are emerging as cost-effective alternatives within healthcare systems. Addressing various challenges in the clinical management of inflammatory bowel disease (IBD) remains crucial. To shed light on physicians’ current knowledge, beliefs, practical approaches, and concerns related to biosimilar adoption—whether initiating a biosimilar, transitioning from an originator to a biosimilar, or switching between biosimilars (including multiple switches and reverse switching)—a global survey was conducted. Fifteen physicians with expertise in the field of IBD from 13 countries attended a virtual international consensus meeting to develop practical guidance regarding biosimilar adoption worldwide, considering the survey results. This consensus centered on 10 key statements covering biosimilar effectiveness, safety, indications, rationale, multiple switches, therapeutic drug monitoring of biosimilars, non-medical switching, and future perspectives. Ultimately, the consensus affirmed that biosimilars are equally effective and safe when compared to originator drugs. They are considered suitable for both biologic-naïve patients and those who have previously been treated with originator drugs, with cost reduction being the primary motivation for transitioning from an originator drug to a biosimilar.

## 1. Introduction

Biological drugs have transformed the treatment of patients with inflammatory bowel disease (IBD), offering a wide range of therapeutic options [1]. However, they have also had a significant economic impact on healthcare budgets, being the primary cost associated with IBD management [2].

The Biologics Price Competition and Innovation Act of 2009 was enacted to promote competition and cost containment across medicine by creating a shortened regulatory pathway for biological products highly similar to licensed biologics (i.e., biosimilars) [3]. Although biologic originator (i.e., bio-originator) patents are expiring, multiple biosimilars are emerging to take their place, offering significant cost-saving opportunities to healthcare systems [4].

From a regulatory standpoint, a robust legal framework has been put in place, and ever since the first biosimilar received approval, the European Medicines Agency (EMA) has implemented a specific pathway for biosimilar approval. This pathway relies on a thorough comparability exercise to ascertain biosimilarity [5,6]. Biosimilars undergo testing for at least one indication of the originator in phase I and III clinical trials, and the results may be extrapolated to other indications [7,8]. Since the initial authorization of a biosimilar by the Food and Drug Administration (FDA) in the United States in 2015, there has been a substantial rise in the number of biosimilars within the field of gastroenterology, with no fewer than 41 biosimilar drugs gaining approval up to the present time.

Certainly, numerous gastroenterologists are currently or will soon have the opportunity to prescribe biosimilars or be tasked with substituting them with previously relied-upon bio-originator drugs. Despite the increasing availability of biosimilars, there remains uncertainty regarding the extent of gastroenterologists’ knowledge and comfort with these products and the practical implications involved in their management [9]. Uptake has been dampened by several issues, including (1) gastroenterologists’ uncertainty about whether safety and efficacy evidence supports interchangeability with bio-originators, (2) the complexity and dynamic nature of reverse and multiple switches, and (3) potential patient acceptance of using a biosimilar and associated nocebo effects [10,11,12].

A 2017 survey of 1201 international physicians from various specialties, including gastroenterology, indicated a need for knowledge-based education about biosimilars [13]. This research highlighted specific knowledge gaps related to biosimilars, such as the inability to correctly define terms like extrapolation and interchangeability in the context of biosimilar regulation and a poor ability to articulate the differences between biosimilars and originators. Thus, it is critically important to better understand gastroenterologists’ attitudes towards biosimilars and their practical management when prescribing these drugs, as these factors affect biosimilar uptake and, ultimately, influence patient outcomes. We surveyed IBD physicians to get their feedback about their management and the current challenges they face when starting a biosimilar, switching from an originator to a biosimilar, or switching from one biosimilar to another (reverse and multiple switching). Furthermore, we organized an international expert consensus meeting to develop and validate evidence-based statements that can serve as guidance for biosimilar use in IBD.

## 2. Methods

### 2.1. Survey

We designed a cross-sectional survey targeted at clinicians involved in the care of IBD worldwide. From January to February 2023, participants were invited to complete a survey hosted on an online platform. The survey link was sent via email through the mailing lists of IBD-scope, a webinar platform designed for healthcare professionals interested in IBD, and through personal invitations to physicians specializing in IBD. Screening questions were included at the beginning of the survey to filter respondents out of the target population, and email registration was used to prevent double participation. Only prescribers were included, regardless of their specialty. The responses were collected anonymously, and permission for data collection was requested at the start of the survey. The survey and the invitation email were in English, and all the questions were multiple-choice.

The questionnaire comprised 46 questions classified into 5 sections (Table 1): the first section recorded participant demographics, specialty, and level of experience (6 questions); the second section focused on practices and attitudes toward biosimilar use in clinical practice (22 questions); the third section dealt with interchangeability, including reverse and multiple switches (9 questions); the fourth section addressed the nocebo effect and non-medical switch (5); and the last section covered current and future perspectives (4 questions). Only the demographic questions were mandatory, and the number of respondents was reported for each question to account for missing data. The full survey is provided in the Appendix A.

The study was conducted and reported in compliance with the Consensus for Reporting of Survey Studies (CROSS) guidelines [14]. The CROSS checklist is available in the Appendix A.

Excel (v16.71, Microsoft) was used to perform descriptive statistics and to plot the charts.

### 2.2. Consensus Meeting

A virtual consensus meeting, including 15 experts in the field of IBD from 13 countries worldwide (Argentina, Brazil, Canada, France, Germany, Ireland, Italy, Japan, Portugal, Spain, Sweden, United Arab Emirates, and United States of America), was held on 21 February 2023. Based on the results of the global survey, 10 statements were formulated by 3 authors (FD, LPB, and SD) and were anonymously voted on by all the experts using Delphi consensus methodology through a virtual platform (Appendix A).

The statements received approval if at least 75% of the participants reached a consensus. In cases where agreement was not initially reached, the statement underwent discussion, rewording, and a second round of voting. If consensus remained elusive in the second round, the statement was excluded. Additionally, new statements could be generated during the virtual meeting, which were subjected to discussion and a vote. All experts participated in the manuscript’s drafting and endorsed its final version.

## 3. Results

### 3.1. Survey

#### 3.1.1. Part 1. Participant Demographics, Specialty, and Level of Experience

In total, 234 physicians from 38 countries across all continents responded to the survey. A total of 39% (99/234) were 30–40 years old, and female and male practitioners were fairly evenly represented in this population (102/234, 43.6% female) (Figure 1). Approximately 86% (202/234) were gastroenterologists, 2.5% (6/234) were surgeons, and the remaining participants were internal medicine specialists (3/254), general practitioners (3/254), and other specialists (20/254). Most participants (174/234, 74.4%) were highly experienced, with more than 10 years of practice in the field of IBD. Regarding center size, 37.1% (87/234) worked in hospitals caring for >500 IBD patients per year, 38.5% (90/234) in institutions caring for 100–500 patients, and 24.3% (57/234) in centers with <100 patients.

#### 3.1.2. Part 2. Practices and Attitudes toward the Use of Biosimilars in Clinical Practice

Most participants (195/234, 83.3%) believed biosimilars to be as effective and safe as originator drugs, and only 3.9% (9/234) stated that they are less effective and safe.

The largest proportion rated their overall confidence with biosimilars as moderate (an average of 8.5 on a scale from 0 to 10). However, participants rated their patients’ average rating of confidence as slightly lower (7.7 vs. 8.5).

Regarding extrapolation, approximately 75% (176/234) of the sample responded that biosimilar data from other immune-mediated inflammatory diseases (IMID) are valid and applicable to the IBD field.

When asked about their practical management, a high proportion of practitioners (195/234, 83.3%) reported that patients should be informed before switching and that physicians, more than nurses, pharmacists, government authorities, or residents, should deliver information. Although more than 70% (165/234) of participants tend to explain to patients what a biosimilar drug is and what an originator drug is, less than a third reported providing data and written material about biosimilars.

Almost all participants (>80%) were familiar with infliximab and adalimumab biosimilars, and they cited lower costs and the unavailability of originators as the main reasons behind biosimilar uptake (54.7% and 15.8%, respectively). Of the practitioners, 72.2% (169/234) believed that biosimilars should not be prescribed only to bio-naïve patients, and the majority thought that switching is a non-medical decision that is independent of patients’ outcomes (e.g., clinical, biochemical, and endoscopic remission). Therapeutic drug monitoring (TDM) with biosimilar trough levels and auto-antibody measurements were performed by 43.2% (94/218) of the participants. Almost half of the practitioners experienced patient refusal to switch, but overall, less than 5% of patients under their care were non-accepting. According to the physicians, lower effectiveness (79/218, 36.2%) and disease flares (52/218, 23.8%), rather than safety concerns (17/218, 7.8%), were indicated as the main sources of fear for patients.

More than half of the practitioners (133/218, 61.0%) still prescribe originators despite the availability of biosimilars. When asked why, the explanations were extremely heterogeneous, with more than 50% not selecting any of the proposed options (Figure 2). Of note, 17.4% (38/218) believed that there is little data available on biosimilars, and 11.9% (26/218) of respondents had no access to biosimilars in their hospital.

#### 3.1.3. Part 3. Interchangeability (Reverse and Multiple Switches)

Approximately one-quarter of participants (57/218, 26.1%) switched from a biosimilar to an originator at least once. The main reasons for the reverse switch were biosimilar unavailability (20/54, 37.0%), the loss of a response to biosimilars (15/54, 27.8%), and allergic reactions to biosimilars (8/54, 14.8%).

When asked “In a patient treated with the originator drug and then switched to the biosimilar, have you ever prescribed the reverse switch to the originator drug?”, almost 40% of the participants responded affirmatively. The decision regarding this modality of reverse switch was based on prior allergic reactions to biosimilars (19/72, 26.4%), the loss of a response to biosimilars (19/72, 26.4%), and unavailability of the first biosimilar (24/72 33.3%). In this specific scenario, most patients (68/72, 94.4%) achieved or maintained disease remission.

Over half of the respondents (93/184, 50.5%) switched from one biosimilar to another biosimilar of the same drug (multiple switches). Consistently with reverse switches, multiple switches led to disease remission and maintained remission in almost all cases. Among the 50.5% of respondents that have experienced multiple switches, the top reasons included first biosimilar unavailability (60/93, 64.5%), allergic reactions (7/93, 7.5%), and the loss of a response (6/93, 6.4%).

#### 3.1.4. Part 4. Nocebo Effect and Non-Medical Switch

Most physicians (159/184, 86.4%) were familiar with the concept of the nocebo effect, but approximately one-third could not confirm whether their patients had ever experienced it. In addition, when physicians were aware of the nocebo effect in their patients, in most cases, they believed it affected less than 5% of them. More than half of IBD specialists (105/184, 57.1%) had patients undergoing non-medical switching, and among them, almost all (98/105, 93.3%) achieved or maintained clinical remission.

#### 3.1.5. Part 5. Current and Future Perspectives

The availability of biosimilars influences therapeutic choices, according to many healthcare practitioners (136/184, 73.9%). Most (96/184, 52.2%) believed they would soon be prescribing biosimilars of vedolizumab and ustekinumab, as well as generics of small molecules. However, a relevant percentage (56/184, 30.4%) stated that the availability of vedolizumab, ustekinumab, and small molecule drugs, such as tofacitinib, would not change the treatment algorithm for IBD patients. When asked for suggestions on how to implement the use of biosimilars in clinical practice, the need for long-term data was pointed out by the majority of respondents (93/184, 50.5%), followed by the need for randomized clinical trials to compare the efficacy and safety of biosimilars and originator drugs in patients with IBD (71/184, 38.6%), more information provided to patients by government authorities (60/184, 32.6%), and greater patient engagement in therapeutic decision-making (59/184, 32.1%) (Figure 3). Information delivered from patient associations and non-medical switching were also indicated by almost a third of respondents.

### 3.2. Consensus Statements

Eight preliminary statements were approved in the first voting round. A list of the preliminary statements is included in the Appendix A. Two statements were approved after the second round of voting. One was not approved during the second voting and was removed, ultimately leading to the approval of ten statements (Table 2).

A consensus was reached for 10 out of the 11 statements (90.1%), with 100% agreement for eight statements (80.0%).

## 4. Discussion

This consensus endorses evidence from clinical trials and increasing experience from observational studies.

Based on these data, biosimilars are equivalent to reference products in terms of their efficacy and safety profiles, including the extrapolation of their indications [15,16].

Regarding patient eligibility for biosimilar switching, both biologic-naïve patients and those already treated with originator drugs are candidates for biosimilar switching. Certainly, the use of biosimilars in naïve patients is better accepted [17]. Indeed, patients who have responded to originator drugs may fear losing their response and, consequently, may experience the nocebo effect. To overcome this limitation, providing adequate patient information is essential and has been associated with significant improvements in patient outcomes [18].

Statement 3 highlighted cost-effectiveness as a primary reason for switching from originator drugs to biosimilars. Given that the high cost of biologics often limits patient access and healthcare system sustainability, biosimilars offer a solution to reduce healthcare expenditure whilst maintaining treatment efficacy and safety.

To date, no study has been specifically designed to evaluate the optimal timing for biosimilar switching. However, based on the reliable efficacy and safety profile of biosimilars and their advantageous cost savings, switching to a biosimilar can be performed at any time.

Statement 6 relied on the concept of interchangeability. Interchangeability allows a biosimilar product to be replaced by the reference product [19,20], making multiple switches from one biosimilar to another feasible in cases of drug unavailability [21,22,23].

Conversely, there is limited evidence for the efficacy of switching from one biosimilar to another in case of a loss of response. For this reason, it should not be recommended. The only rejected statement was the one concerning reverse switching from a biosimilar to an originator drug. A Dutch retrospective multicenter study revealed that reverse switching is performed in about 10% of cases due to worsening gastrointestinal symptoms, adverse effects, or a loss of response [24]. Interestingly, approximately three-quarters of patients who lost their response after switching to a biosimilar regained their response after reverse switching to an originator. This observation is intriguing and suggests that there may be situations where this strategy could be considered. However, the lack of robust prospective studies means that this approach cannot be widely recommended in clinical practice at this time. Further research is needed to clarify the circumstances under which a reverse switch may be a viable option and to better understand the factors influencing individual patient responses to biosimilars and reference biologics.

Another crucial aspect regarding the switch to biosimilars is the need to monitor drug levels and anti-drug antibodies. There is substantial evidence for the TDM of biologic therapies in IBD. Notably, there are no specific TDM assays for biosimilars. In addition, there are no data demonstrating a higher risk of immunogenicity with biosimilars. Therefore, there is no need to modify the regular practice of monitoring drug trough levels and anti-drug antibodies in patients that have been switched from an originator drug to a biosimilar [25,26,27]. TDM should not be encouraged for patients solely because of switching to a biosimilar when they are in remission. Despite the notable advantages of the use of biosimilars, there are still patients who refuse the switch and prescribers who still prefer originator drugs.

The non-medical switch (i.e., a change to a patient’s medication for reasons other than the lack of a clinical response, safety issues, or poor compliance) could be the key to reducing the costs associated with advanced therapies and increasing accessibility. A prospective multicenter study showed no differences in efficacy and safety between patients treated with originator drugs and those who switched to biosimilars for non-medical reasons [28]. Similarly, a systematic review found no effectiveness, safety, or immunogenicity concerns with non-medical switching, supporting its wide use [15]. Currently, only biosimilars of infliximab and adalimumab are available. Of note, the patents on vedolizumab and ustekinumab will soon expire, expanding the range of biosimilars [29,30]. A recent randomized clinical trial enrolling healthy volunteers compared the safety and immunogenicity of ustekinumab and its biosimilar [31]. No difference in pharmacokinetics was identified. The safety profile of the two drugs was also comparable. Non-anti-TNF biosimilars are expected to become available in the foreseeable future, potentially altering therapeutic algorithms for patients with IBD. TNF antagonists are currently the first therapeutic option for most patients due to their efficacy, known safety, and low cost.

Other biological drugs and small molecules are frequently used as subsequent lines of therapy. There are exceptions in case of contraindications to TNF inhibitors or preferences for another drug in the management of specific subpopulations (e.g., extraintestinal manifestations, the elderly, co-morbidities, perianal disease, or chronic pouchitis) [32,33]. The non-anti-TNF biosimilars, once approved by regulatory authorities, will allow further reductions in healthcare costs. Moreover, an increasingly personalized and tailored approach will be evaluated, selecting the most suitable drug for the individual patient regardless of the economic impact.

We acknowledge some important limitations. Firstly, the questionnaire was not completed by all the respondents. However, the number of respondents for each question was reported to overcome missing data. Furthermore, participants came from different countries around the world with different local regulatory authorities, which may have contributed to the heterogeneity of responses.

## 5. Conclusions

Biologic therapies have brought about substantial enhancements in the outcomes of individuals with IBD, but they have also resulted in increased healthcare expenses [34]. The signature promises of biosimilars are to improve patient access to highly effective treatments earlier in the disease course, to decrease costs, and to maintain improved health outcomes. This represents an opportunity for gastroenterologists to deliver high-quality and high-value care. The present survey provides insights regarding IBD specialists’ practical management, beliefs, and knowledge surrounding biosimilars. The subsequent International Delphi provides the first consensus statements for the use of biosimilars in the treatment of patients with IBD, affirming that biosimilars are equally effective and safe compared to originator drugs. They are suitable for both biologic-naïve patients and those previously treated with originator drugs, with cost reduction being the primary motivation for transitioning from an originator drug to a biosimilar.

These consensus statements are intended to offer guidance to clinicians, healthcare organizations, the pharmaceutical industry, and patients regarding the development and use of biosimilars for the treatment of IBD around the world.

## Figures and Tables

**Figure 1 jcm-12-06350-f001:**
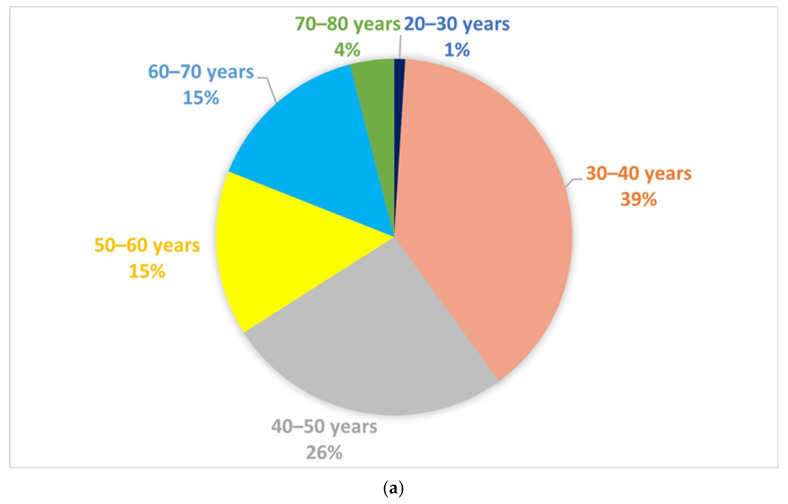
Characteristics of survey respondents: (**a**) age; (**b**) sex; and (**c**) geographic representation.

**Figure 2 jcm-12-06350-f002:**
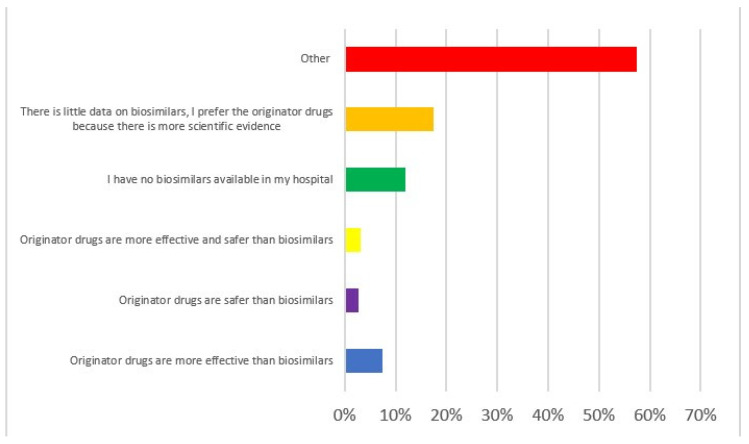
Replies to Q27 “If you prescribe originator drugs despite the availability of the biosimilars, why do you prefer the originator drug?”.

**Figure 3 jcm-12-06350-f003:**
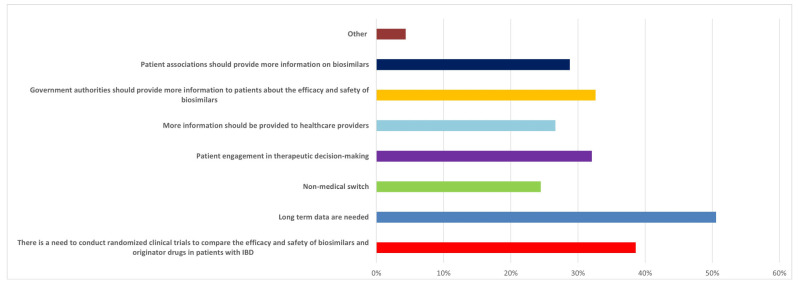
Replies to Q46. “How would you implement the use of biosimilars in clinical practice (multiple answers are possible)?”.

**Table 1 jcm-12-06350-t001:** The survey and its 5 domains.

**Part 1 Participant demographics, specialty, and level of experience** Q1. Age in years Q2. Sex Q3. What country do you work in? Q4. What is your specialization? Q5. How many years of experience do you have in the field of IBD? Q6. How many IBD patients do you see per year? **Part 2 Practices and attitudes toward the use of biosimilars in clinical practice** Q7. In your opinion, which of the following statements is correct? Q8. How confident are you about the use of biosimilars using a scale from 0 (lowest value) to 10 (highest value)? Q9. How confident are your patients about the use of biosimilars using a scale from 0 (lowest value) to 10 (highest value)? Q10. Do you think the data on biosimilars extrapolated from other immune-mediated inflammatory diseases are also valid in IBD? Q11. Do you think patients should be informed about what a biosimilar is before starting therapy? Q12. Who should provide information to patients about biosimilars? Q13. Before prescribing a biological drug, do you explain to patients what a biosimilar drug is and what the originator drug is? Q14. Do you provide patients with data comparing biosimilars and originator drugs? Q15. Do you provide written materials to patients informing them on the use of biosimilars? Q16. Have you ever prescribed biosimilars of infliximab? Q17. Have you ever prescribed biosimilars of adalimumab? Q18. Do you think biosimilars should only be prescribed to naïve patients? Q19. Have you ever switched a patient from an originator drug to a biosimilar drug? Q20. If you answered yes to question 19, why did you switch from an originator drug to a biosimilar drug? Q21. When do you switch from an originator to a biosimilar? Q22. Do you monitor drug trough levels and auto-antibodies in patients switched to biosimilars? Q23. Do you have patients who have refused to start therapy with a biosimilar or switch to a biosimilar? Q24. If you have patients who refused to be treated with a biosimilar, what is the proportion of these patients? Q25. What is the main reason for patients’ refusal of biosimilar drugs? Q26. Despite the availability of biosimilars, do you prescribe originator drugs? Q27. If you prescribe originator drugs despite the availability of the biosimilars, why do you prefer the originator drug? Q28. In a patient candidate for biologic therapy, have you ever started IBD therapy using biosimilars? **Part 3. Interchangeability (reverse and multiple switch)** Q29. In a patient who started on a biosimilar as their first drug, have you ever switched them to an originator drug? Q30. If you answered yes to question 29, why were the patients switched from a biosimilar to an originator drug (multiple answers are possible)? Q31. If you answered yes to question 29, did the patients who were switched from a biosimilar to an originator drug achieve/maintain disease remission? Q32. Have you ever switched from one biosimilar to another biosimilar of the same drug (multiple switch)? Q33. If you have multiple switched patients, why were the patients switched from one biosimilar to another (multiple answers are possible)? Q34. Did multiple switched patients achieve/maintain disease remission? Q35. In a patient treated with the originator drug and then switched to the biosimilar, have you ever prescribed the reverse switch to the originator drug? Q36. If you have patients undergoing reverse switch, what was the reason for reverse switch (multiple answers are possible)? Q37. Did reverse switched patients achieve/maintain disease remission? **Part 4. Nocebo effect and non-medical switch** Q38. Do you know what the nocebo effect is? Q39. Have your patients ever experienced the nocebo effect? Q40. If your patients experienced the nocebo effect, what is the rate of nocebo effects among your patients? Q41. Do you have patients who underwent a non-medical switch? Q42. Did non-medical switched patients achieve/maintain disease remission? **Part 5. Current and future perspectives** Q43. Does the presence of biosimilars have an impact on your therapeutic choices? Q44. In the near future, will you be prescribing biosimilars of vedolizumab, ustekinumab, and tofacitinib? Q45. Do you think the availability of the biosimilars of vedolizumab, ustekinumab, and tofacitinib will change the treatment algorithm of IBD patients? Q46. How would you implement the use of biosimilars in clinical practice (multiple answers are possible)?

**Table 2 jcm-12-06350-t002:** Approved statements and agreement after the second round of voting. * after the second round of voting.

Statements		Agreement >75% (%)
1	Biosimilars are as effective and safe as originator drugs.	100%
2	Biosimilars can be used both in biologic-naïve patients and in patients already treated with originator drugs.	100%
3	The main reason for switching from an originator drug to a biosimilar is its lower cost.	100%
4	Switching from an originator drug to a biosimilar can be performed at any time.	82%
5	The switch from an originator drug to a biosimilar is effective and safe.	100%
6	Multiple switches from one biosimilar to another are feasible in case of drug unavailability.	100%
7	We do not recommend multiple switches in case of the loss of a response to a biosimilar.	100% *
8	There is no need to modify regular practice in monitoring drug trough levels and antibodies in patients switched from originator drugs to biosimilars.	90% *
9	Non-medical switching is a way to reduce the costs associated with advanced therapies and to increase accessibility.	100%
10	In the near future, non-anti-TNF biosimilar drugs are expected to alter the therapeutic algorithm for patients with inflammatory bowel disease.	100%

## Data Availability

Not applicable.

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
