# Peer review of "Practical Management of Biosimilar Use in Inflammatory Bowel Disease (IBD): A Global Survey and an International Delphi Consensus"

_jcm, 2023, doi:10.3390/jcm12196350_

Round 1

Reviewer 1 Report

I think that this Manuscript covers a very important topic many clinicians will be facing in the future when they need to weight whether a patient should be switched to a biosimilar and savings can be made or patients concerns should come first. However, this study included a rather small number of participants. Can authors provide rationale for the sample size or response rate data if possible? because I believe that the small sample questions the validity of the findings. Also, it should be pointed out this is a mixed methods study.

I would like to see more discussion on this finding loss of response (15/54, 27.8%), it would be interesting to investigate how often clinicians report drug failure to the agencies as this questions the science behind biosimilars

Figure 3 needs better resolution

Most participants (195/234, 83.3%) believed biosimilars are as effective and safe as 144 the originator drugs, this sentence does not align with data in Table 2, please clarify if data in table 2 is an onwards consensus of only a few experts

A part of the results, especially 3.2. section should be placed in discussion section as discussion is oddly concise.

Author Response

Comments from reviewer 1

I think that this Manuscript covers a very important topic many clinicians will be facing in the future when they need to weight whether a patient should be switched to a biosimilar and savings can be made or patients concerns should come first.

Reply: We thank the reviewer for the positive comment.

However, this study included a rather small number of participants.

Can authors provide rationale for the sample size or response rate data if possible? Because I believe that the small sample questions the validity of the findings. Also, it should be pointed out this is a mixed methods study.

Reply: No formal sample size calculation was conducted. We planned on inviting clinicians involved in the care of IBD worldwide, using an online platform. The link was sent via email through the mailing lists of IBD‐scope, a webinar platform designed for healthcare professionals interested in IBD, and by personal invitation of physicians with a focus on IBD. We anticipated inviting approximately 400 subjects, with an expected questionnaire response rate of 30-50%, aligned with the Diliman Total Design Survey Method.

I would like to see more discussion on this finding loss of response (15/54, 27.8%), it would be interesting to investigate how often clinicians report drug failure to the agencies as these questions the science behind biosimilars.

Reply: We agree with the reviewer and we added this interesting point in the discussion. The observation that many patients regained response after a reverse switch to the originator drug is intriguing and suggests that there may be situations where this strategy could be considered. However, the lack of robust prospective studies means that this approach cannot be widely recommended in clinical practice at this time. Further research is needed to clarify the circumstances under which a reverse switch may be a viable option and to better understand the factors influencing individual patient responses to biosimilars and reference biologics.

Figure 3 needs better resolution.

Reply: We modified Figure 3 as required by reviewer 3.

Most participants (195/234, 83.3%) believed biosimilars are as effective and safe as 144 the originator drugs, this sentence does not align with data in Table 2, please clarify if data in table 2 is an onwards consensus of only a few experts.

Reply: We thank the reviewer for the comment. It is indeed as he/she correctly said. We made it clearer.

A part of the results, especially 3.2. section should be placed in discussion section as discussion is oddly concise.

Reply: We modified Results and Discussion sections as required.

Reviewer 2 Report

The authors of this study have applied a global survey and also a consensus on practical management of biosimilar use in inflammatory bowel disease (IBD). This title is worthy of investigation. However, regarding reviewing the manuscript, some comments are mentioned below that need more clarification, explanation, or modification.

Abstract:

1.       It is better that abstract be structured and divided into background, objectives, methods, results, and conclusion.

2.       It is better to mention the results or a brief conclusion of the study in abstract.

3.       Why irritable bowel disease (IBD) is not among the keywords of the study? Although there are Crohn’s disease and Ulcerative colitis among the keywords, it is better to mention IBD too.

Introduction:

1.       IBD that was utilized in line 82 should first be mentioned in its complete form.

2.       IBD treatments with bio-similar or bio-originator drugs and their efficacy were not discussed in introduction. Although IBD and IBD management is one of the main topics of this manuscript, it was not described except in two sentences at the end of introduction. Please reconsider it.

Methods:

1.       Please mention the 10 statements that were voted by participants and the criteria utilized in selecting these statements.

2.       Please mention the inclusion and exclusion criteria of the study (including participants, etc.).

3.       How were the 46 questioned selected? Did you follow or imitate a specific study or not?

Results:

1.       Type of the charts that were utilized in figure 1 and figure 3 are not suitable. Please change them in a way that is more practical and also perceivable with a glance.

Discussion:

1.       You should name the section that is started from line 239 to 295, Discussion section.

2.       The name of the section that was started from line 297 to 307 should be changed to Conclusion.

3.       Please mention the limitations of the study at the end of discussion.

4.       Generally the discussion is not well organized. Paragraphs are too long. Besides, several topics are discussed in a single paragraph. It is better to mention every single finding of the study in a single paragraph and afterwards compare this finding with the current literature.

Wish you luck and prosperity in preparing the manuscript.

Minor editing of English language is required. 

Author Response

Comments from reviewer 2:

The authors of this study have applied a global survey and also a consensus on practical management of biosimilar use in inflammatory bowel disease (IBD). This title is worthy of investigation. However, regarding reviewing the manuscript, some comments are mentioned below that need more clarification, explanation, or modification.

Abstract:

  1. It is better that abstract be structured and divided into background, objectives, methods, results, and conclusion.

  1. It is better to mention the results or a brief conclusion of the study in abstract.

Reply: We thank the reviewer for the comments above. According to JCM guidelines, “the abstract should be a total of about 200 words maximum. The abstract should be a single paragraph and should follow the style of structured abstracts, but without headings”. We aligned with the JCM guidelines for the layout but we mentioned results and conclusion in the abstract.

  1. Why irritable bowel disease (IBD) is not among the keywords of the study? Although there are Crohn’s disease and Ulcerative colitis among the keywords, it is better to mention IBD too.

Reply: We thank you the reviewer and we made the commended changes.

Introduction:

  1. IBD that was utilized in line 82 should first be mentioned in its complete form.

Reply: We made the recommended changes

  1. IBD treatments with bio-similar or bio-originator drugs and their efficacy were not discussed in introduction. Although IBD and IBD management is one of the main topics of this manuscript, it was not described except in two sentences at the end of introduction. Please reconsider it.

Reply: We thank the reviewer and we add a sentence in the introduction.

Methods:

  1. Please mention the 10 statements that were voted by participants and the criteria utilized in selecting these statements.

Reply: We have added the 10 preliminary statements as supplement data. Statements were chosen based on the relevance of the survey results as clarified in the methods.

  1. Please mention the inclusion and exclusion criteria of the study (including participants, etc.).

Reply: We have added the inclusion criteria as recommended.

  1. How were the 46 questioned selected? Did you follow or imitate a specific study or not?

Reply: We thank the reviewer for the comment. No other questionnaire was used as an example. The questions were formulated on the basis of daily clinical practice to cover all possible scenarios.

Results:

  1. Type of the charts that were utilized in figure 1 and figure 3 are not suitable. Please change them in a way that is more practical and also perceivable with a glance.

Reply: We thank the reviewer for the comment. We modified figure 3 in order to make more practical. Demographics should be reflected as pie chart in order to make it clear the role of diversity and inclusion among participants.

Discussion:

  1. You should name the section that is started from line 239 to 295, Discussion section.

Reply: We made the recommended changes.

  1. The name of the section that was started from line 297 to 307 should be changed to Conclusion.

Reply: We made the recommended changes

  1. Please mention the limitations of the study at the end of discussion.

Reply: We have mentioned the limitations of the study at the end of discussion as requested.

  1. Generally the discussion is not well organized. Paragraphs are too long. Besides, several topics are discussed in a single paragraph. It is better to mention every single finding of the study in a single paragraph and afterwards compare this finding with the current literature.

Reply: We thank the reviewer for this comment. We have shortened the paragraphs as recommended. All topics are addressed separately.

Reviewer 3 Report

In the manuscript “Practical management of biosimilar use in inflammatory bowel disease (IBD): a global survey and an International Delphi consensus”, the authors present and discuss the observed results from a survey with physicians (mostly gastroenterologists) from several countries and with different levels of age and experience on the management of patients with IBD.

Summarizing these different perspectives from around the globe and reaching an expert consensus about the use of biosimilars in IBD is a major step towards the guidance to clinicians, healthcare organizations, pharmaceutical industry, and patients regarding the development and use of biosimilars for the treatment of IBD. This will be an important addition to the literature as long as the authors correct the excessive number of self-citations, which is >20% of the references. Close to 50% of the references at the introduction are self-citations.

Author Response

Comments from reviewer 3:

In the manuscript “Practical management of biosimilar use in inflammatory bowel disease (IBD): a global survey and an International Delphi consensus”, the authors present and discuss the observed results from a survey with physicians (mostly gastroenterologists) from several countries and with different levels of age and experience on the management of patients with IBD.

Summarizing these different perspectives from around the globe and reaching an expert consensus about the use of biosimilars in IBD is a major step towards the guidance to clinicians, healthcare organizations, pharmaceutical industry, and patients regarding the development and use of biosimilars for the treatment of IBD.

This will be an important addition to the literature as long as the authors correct the excessive number of self-citations, which is >20% of the references. Close to 50% of the references at the introduction are self-citations.

Reply: We thank the reviewer for the comments and we reduced the number of self-citations. We included only 3 references from the overall 15 authors (8.8%).

Round 2

Reviewer 2 Report

Thanks for the revision of the manuscript.

The manuscript is much more better now. Almost all of the comments were responded properly and modification was done in a good order. However, still the discussion part can be reorganized better. In addition, it was better to present the key findings of the study briefly in the conclusion part not just the methods of the study. 

Wish you the best in preparing the manuscript. 

Author Response

We implemented the discussion to make it aligned with the statements list and order. We added a summary of the results in the conclusion section.

Reviewer 3 Report

Congratulations to the team on this interesting manuscript.

Author Response

We thank the reviewer for this positive comment